# Compacter: A Lightweight Transformer for Image Restoration

Zhijian Wu
East China Normal University
Shanghai, China
zjwu_97@stu.ecnu.edu.cn

Jun Li
Nanjing Normal University
Nanjing, China
lijuncst@njnu.edu.cn

Yang Hu
New York University
Brooklyn, USA
yh5965@nyu.edu

Dingjiang Huang*
East China Normal University
Shanghai, China
djhuang@dase.ecnu.edu.cn

## Abstract

Although deep learning-based methods have made significant advances in the field of image restoration (IR), they often suffer from excessive model parameters. To tackle this problem, this work proposes a compact Transformer (Compacter) for lightweight image restoration by making several key designs. We employ the concepts of *projection sharing*, *adaptive interaction*, and *heterogeneous aggregation* to develop a novel Compact Adaptive Self-Attention (CASA). Specifically, CASA utilizes shared projection to generate Query, Key, and Value to simultaneously model spatial and channel-wise self-attention. The adaptive interaction process is then used to propagate and integrate global information from two different dimensions, thus enabling omnidirectional relational interaction. Finally, a depth-wise convolution is incorporated on Value to complement heterogeneous local information, enabling global-local coupling. Moreover, we propose a Dual Selective Gated Module (DSGM) to dynamically encapsulate the globality into each pixel for context-adaptive aggregation. Extensive experiments demonstrate that our Compacter achieves state-of-the-art performance for a variety of lightweight IR tasks with approximately 400K parameters.

## CCS Concepts

• **Computing methodologies** → **Computer vision**; **Reconstruction**; • **Computer systems organization** → **Neural networks**.

## Keywords

Deep Learning, Lightweight Image Restoration, Self-Attention

**ACM Reference Format:**

Zhijian Wu, Jun Li, Yang Hu, and Dingjiang Huang. 2024. Compacter: A Lightweight Transformer for Image Restoration. In *Proceedings of the 32nd ACM International Conference on Multimedia (MM '24), October 28-November 1, 2024, Melbourne, VIC, Australia.* ACM, New York, NY, USA, 10 pages. https://doi.org/10.1145/3664647.3680811

---

*Corresponding author.

---

## 1 Introduction

As a classical task in low-level computer vision, image restoration (IR) aims to recover high-quality counterparts from degraded images by removing degraded content. With the rapid development of deep learning, convolutional neural networks (CNNs) have become the de facto method for modern IR algorithms. Various advanced CNN designs were introduced into the image restoration task leading to further performance progress [55, 61, 62]. However, CNNs suffer from limitations in long-range dependencies modeling because convolution operators are more advantageous in extracting local information [21]. Recently, the dominance of CNNs has been challenged by vision Transformer-based models, which exhibit superior performance due to the advantage of self-attention in long-range modeling [15]. However, the quadratic complexity of self-attention makes it difficult to apply to image restoration tasks involving high-resolution images [6]. Many attempts have been made to develop more efficient self-attention mechanisms for image restoration. SwinIR [26] uses the shifted window mechanism to limit the scope of attention to the window, such that linear complexity is achieved. Similar strategies were adopted by Uformer [42] for building a hierarchical network architecture. Restormer [53] proposes to use channel-wise self-attention instead of spatial one to achieve linear complexity.

Despite significant advances, most attempts have been devoted to improving large-scale image restoration models while neglecting the development of lightweight models. Lightweight image restoration networks are still fraught with great challenges. Specifically, most existing methods stack a single spatial or channel self-attention, making single-dimensional information modeling fail to achieve comprehensive feature interactions. Although some recent methods [7, 37] have been proposed to use the two self-attention mechanisms alternatively, it may lead to a large number of model parameters. In addition, the simple sequential combination is not fully capable of effectively modeling omnidirectional higher-order relational interactions. Besides, self-attention focuses more on global long-range information and is inferior at modeling local information [33], which is detrimental to pixel-level image restoration tasks. These problems are more severe in lightweight models, which cannot stack enough layers due to limited capacity.

In this paper, we propose a novel compact Transformer called Compacter for lightweight image restoration. Specifically, we propose a compact adaptive self-attention (CASA) mechanism through

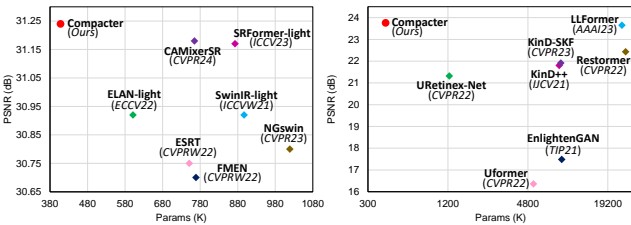

**Figure 1: Comparisons between our Compacter and other state-of-the-art algorithms. *Left*: PSNR vs. Parameters on the Manga109 [32] dataset for super-resolution (×4). *Right*: PSNR vs. Parameters on the LOL [43] dataset for low-light enhancement. The proposed Compacter achieves a better trade-off between performance and parameters.**

the design of *projection sharing*, *adaptive interaction*, and *heterogeneous aggregation*. CASA first utilizes shared projections for generating Query, Key, and Value to simultaneously establish spatial and channel self-attention. Subsequently, the bidirectional interaction process is utilized to adaptively propagate and integrate the two global information from two different axes, thereby realizing omnidirectional relational interactions. Finally, a depth-wise convolution is performed in parallel on Value to complement the locality to achieve heterogeneous aggregation of global and local information. With these designs, CASA enables comprehensive information dissemination and interaction within a compact computational unit. Furthermore, we propose a dual selective gated module (DSGM) to further calibrate the aggregated features and obtain per-pixel global dependencies. DSGM utilizes parallel branches to encode local information and gated to each other to adaptively promote favorable pixels and suppress detrimental pixels, thus producing high-quality restoration results. Thanks to two complementary components, our Compacter enables comprehensive pixel-relational interaction and maintains a desirable model size. We conducted comprehensive experiments on four lightweight image restoration tasks, our method achieved superior performance with significantly fewer parameters. Figure 1 illustrates the comparison of the image super-resolution and low-light enhancement tasks. We further provide extensive ablation experiments to demonstrate the effectiveness of the architectural design.

To sum up, we summarize the contributions of this paper as follows:

- We propose a compact Transformer-based network, called Compacter, for lightweight image restoration.
- We present compact adaptive self-attention (CASA) that achieves adaptive interaction through cross-modulation between spatial and channel global information. A novel dual selective gated module (DSGM) is proposed for dynamic context aggregation.
- Extensive experiments show that Compacter achieves SOTA results on various lightweight restoration tasks with significantly fewer parameters.

## 2 Related Work

### 2.1 Image Restoration

The purpose of image restoration is to recover a clean version from a degraded image. In recent years, deep learning-based image restoration methods have achieved unprecedented success. In particular, CNN models have achieved state-of-the-art performance on various tasks such as super-resolution [16, 29, 38], denoising [54], deraining [23, 49], and low-light enhancement [24, 59]. Image restoration algorithms are driven by the design of advanced CNN network architectures. For example, the residual connection has been introduced for tasks such as super-resolution [62], image denoising [20], and has become a necessary component of image restoration networks. Multi-scale representation learning is also widely used in various restoration tasks [55] due to its superior performance. Subsequently, the emerging attention mechanism has been explored for further performance improvements in low-level vision tasks. RCAN [61] coupled channel attention within the SR network, remarkably improving the representational capability of the model. To better learn feature correlation, SAN [13] further proposed a second-order channel attention module. However, most attempts have been devoted to improving the performance of large restoration models, while neglecting the development of lightweight models. Although some lightweight methods [27, 29, 45] have emerged recently, they tend to be dedicated to specific tasks such as super-resolution. In contrast, our Compacter is a general-purpose lightweight image restoration network that can be used for various restoration-related tasks such as image super-resolution, denoising, deraining, and low-light enhancement.

### 2.2 Vision Transformers

Vision Transformer [15] has achieved impressive performance due to the self-attention mechanism, which naturally incorporates powerful dynamic weights and global dependency capture. However, the quadratic computational complexity of self-attention limits the application to vision tasks, which typically involve high-resolution images. IPT [6] pioneered introducing vision Transformer to image restoration tasks, which reduces the computational cost by decomposing the image into small patches and using a sequence of small patches as input. After that, SwinIR [26] used shifted-window-based self-attention to reduce the computational cost by restricting the scope of self-attention to the local window. A similar strategy is also used by Uformer [42], which constructs a U-shaped hierarchical network. In addition, Restormer [53] adopts channel attention instead of the original spatial attention, making the computational complexity linear in spatial resolution. Despite achieving promising performance, they restrict attention to a local scope and may not fully exploit the potential of Transformers in capturing global dependencies [64]. Some recent methods [7, 37] suggest alternating spatial and channel attention to achieve further performance improvements. However, these serial models do not adequately represent the process of fusion of spatial and channel information, due to the lack of ability to model their interactions. In addition, these efforts remain committed to developing large vision Transformer models. In contrast, this paper is dedicated to designing a lightweight image restoration network.

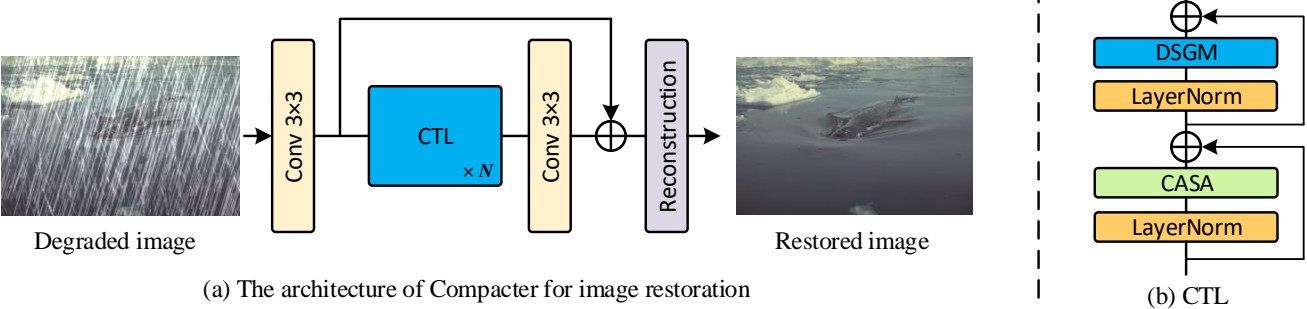

(a) The architecture of Compacter for image restoration

(b) CTL

Figure 2: (a) The overall architecture of our proposed Compacter for image restoration. (b) The inner structure of the compact transformer layer (CTL).

## 3 Method

### 3.1 Overall Architecture

As shown in Figure 2 (a), the overall architecture of our Compacter consists of three main parts: shallow feature extraction, deep feature extraction, and reconstruction module. Specifically, the input low-quality image $I_{LQ}$ is converted using a $3 \times 3$ convolution to extract shallow features. Subsequently, we use deep feature extraction to generate deep features $F_D$. The deep feature extraction consists of $N$ compact transformer layers (CTL) and a $3 \times 3$ convolutional layer with residual concatenation applied. Finally, the deep features are fed to the reconstruction module to construct a high-quality output image $I_{HQ}$. The composition of the reconstruction module depends on different image restoration tasks. For image SR, a sub-pixel convolutional layer [34] is used to reconstruct the high-resolution image $I_{HQ}$. For other image reconstruction tasks, a convolutional layer is used to generate a residual image $I_R$ which is added to the degraded image to obtain the restored image $I_{HQ} = I_{LQ} + I_R$.

**Loss Function.** Following prior works [12, 35], we optimize our model using dual-domain loss:

$$\mathcal{L}_1 = ||I_P - I_G||_1, \quad \mathcal{L}_f = ||\mathcal{F}(I_P) - \mathcal{F}(I_G)||_1, \tag{1}$$

$$\mathcal{L} = \mathcal{L}_1 + \lambda \mathcal{L}_f \tag{2}$$

where $I_P$ and $I_G$ represent the predicted image and ground truth. $\mathcal{F}(\cdot)$ is the 2D Fast Fourier Transform. $\lambda$ is set to 0.1 for balancing dual-domain training.

### 3.2 Compact Transformer Layer

As shown in Figure 2 (b), our Compact Transformer Layer (CTL) consists of a compact adaptive self-attention (CASA) and a dual selective gated module (DSGM). It can be formulated as:

$$X' = \text{CASA}(\text{LN}(X_{in})) + X_{in}, \tag{3}$$

$$X_{out} = \text{DSGM}(\text{LN}(X')) + X' \tag{4}$$

where $X_{in}$ and $X_{out}$ are the input and output features, $LN(\cdot)$ denotes the LayerNorm operation [3].

### 3.3 Compact Adaptive Self-Attention

Comprehensive and dense feature interactions help reconstruct finer results in restoration tasks that require dense per-pixel predictions. To fully mine the potential full correlation of features within a compact computational unit, we propose compact adaptive self-attention (CASA), as shown in Figure 3. Our CASA is based on the philosophy of *projection sharing*, *adaptive interaction* and *heterogeneous aggregation*. It not only adaptively models spatial and channel-wise global information and their interactions but also achieves global-local coupling through the aggregation of heterogeneous operators.

**Projection Sharing.** Given an input feature map $X \in \mathbb{R}^{C \times H \times W}$, we use linear layers to project it into query, key, and value $Q, K, V \in \mathbb{R}^{C \times H \times W}$. Subsequently, the generated $Q, K, V$ are fed to spatial window self-attention (SWSA) and channel-wise self-attention (CWSA), respectively. Based on this design, SWSA and CWSA share $Q, K, V$ generated by the same projection, allowing to modelling of both spatial and channel information in a more compact manner.

$$Q, K, V = \text{Linear}_i(X), i \in \{0, 1, 2\} \tag{5}$$

$$Y_s = \text{SWSA}(Q, K, V), \quad Y_c = \text{CWSA}(Q, K, V) \tag{6}$$

where Linear$(\cdot)$ denotes linear layer. The computational processes of SWSA and CWSA are elaborated as follows.

*Spatial Window Self-Attention (SWSA).* Given the query, key and value $Q, K, V$, we split them into non-overlapping windows with the resolution of $N_w$ and flatten them in $\hat{Q}, \hat{K}, \hat{V} \in \mathbb{R}^{N_w \times \frac{HW}{N_w} \times C}$. The SWSA is formulated as:

$$\hat{Q}, \hat{K}, \hat{V} = \text{partition}(Q, K, V) \tag{7}$$

$$\text{SWSA} = \text{Softmax}(\hat{Q} \cdot K^{\hat{T}}/\sqrt{d} + B) \cdot \hat{V} \tag{8}$$

where $B$ is a dynamic position bias, whose value is learned through an MLP [40].

*Channel-Wise Self-Attention (CWSA).* Given the query, key, and value $Q, K, V$, we reshape them into $\hat{Q}, \hat{K}, \hat{V} \in \mathbb{R}^{C \times HW}$ and compute the transposed-attention map across channels. The CWSA process is defined as:

$$\hat{Q}, \hat{K}, \hat{V} = \text{reshape}(Q, K, V) \tag{9}$$

$$\text{CWSA} = \text{Softmax}(\hat{Q} \cdot K^{\hat{T}}/\alpha) \cdot \hat{V} \tag{10}$$

where $\alpha$ is a learnable scaling factor.

**Adaptive Interaction.** After establishing spatial and channel long-range dependencies, we use a bidirectional interaction process to adaptively propagate and integrate the two global information from two different axes. We adopt the cross-modulation method [17] to

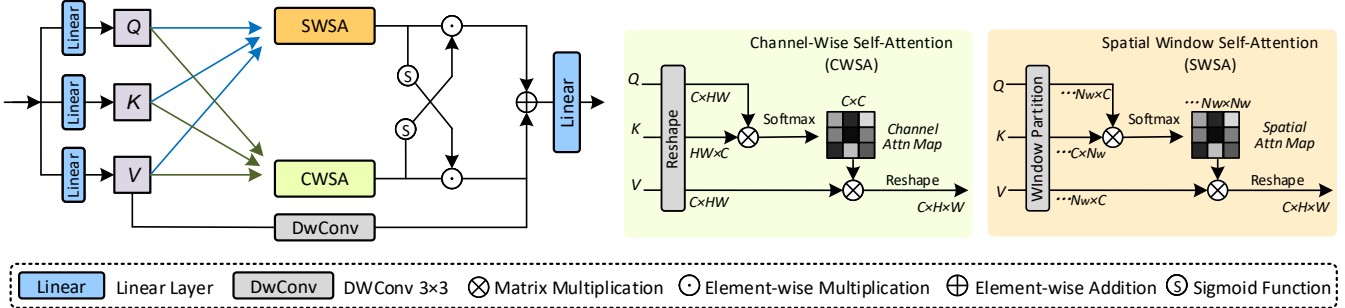

Figure 3: Illustration of compact adaptive self-attention (CASA).

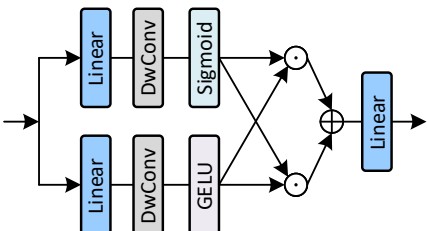

Figure 4: Illustration of dual selective gated module (DSGM).

realize the bidirectional interaction. The computational procedure is formulated as follows:

$$\hat{Y}_s = Y_s \odot \text{Sigmoid}(Y_c), \quad \hat{Y}_c = Y_c \odot \text{Sigmoid}(Y_s) \quad (11)$$

where $\odot$ is element-wise multiplication. With bidirectional interaction, the spatial and channel self-attentions complement and augment each other, thereby enabling omni-axial global representations.

**Heterogeneous Aggregation.** Although self-attention excels at modeling long-range dependencies, it lacks an inductive bias that favors extracting high-frequency local information [21]. To address this issue, we propose to add a depth-wise convolution operation on the value $V$ to complement the heterogeneous local information. Finally, all these representations are combined followed by aggregation using a linear layer:

$$\text{CASA} = \text{Linear}(\hat{Y}_s + \hat{Y}_c + \text{DwConv}(V)) \quad (12)$$

Based on this design, our CASA not only realizes comprehensive dense feature interactions but also provides heterogeneous feature aggregation capability at different granularities within a compact computational unit.

### 3.4 Dual Selective Gated Module

We propose a dual selective gated module (DSGM) to further calibrate the aggregated features generated by CASA and encapsulate the comprehensive feature correlation to each pixel. Similar to the gating mechanism [14], features are fed to two parallel linear layers followed by depth-wise convolution to encode local pixel information.

$$Z_1 = \text{DwConv}(\text{Linear}(X)), \quad Z_2 = \text{DwConv}(\text{Linear}(X)) \quad (13)$$

The local-aware representations are converted into context-aware weights via Sigmoid and GELU activation functions, and adaptively filter the extracted pixel features using dual selective gating. Finally, a linear layer is used to aggregate these representations.

$$\hat{Z}_1 = Z_1 \odot \text{GELU}(Z_2), \quad \hat{Z}_2 = Z_2 \odot \text{Sigmoid}(Z_1) \quad (14)$$

$$DSGM = \text{Linear}(\hat{Z}_1 + \hat{Z}_2) \quad (15)$$

The two different activation functions allow generating weights with different properties to adaptively promote beneficial pixels and suppress detrimental pixels when performing information aggregation. With the above design, our DSGM can more efficiently aggregate global information and local contexts while introducing less computation.

## 4 Experiment

### 4.1 Experimental Setup

*4.1.1 Datasets and Evaluation Metrics.* We evaluate our Compacter on four image processing tasks. For the sake of fairness, the datasets used for all experiments follow exactly previous works [9, 10, 53, 65]. Training and testing datasets for each task are listed below.

**Image Super-Resolution.** Our model is trained on the DIV2K [1] dataset. The evaluation is implemented on five public datasets: Set5 [5], Set14 [48], BSD100 [31], Urban100 [22] and Manga109 [32].

**Image Denoising.** The image denoising task is trained on DIV2K, Flickr2K [36], BSD400 [2], and WED [30], and tested on CBSD68 [31], Kodak [18], McMaster [58], and Urban100 [22],

**Low-Light Image Enhancement.** The training and testing are implemented in the LOL [44] dataset.

**Image Deraining.** We use Rain13K [19] dataset for training, while evaluate on Rain100H [50] and Test100 [56].

**Evaluation metrics.** The common PSNR and SSIM are used to measure performance. Following standard practice, both metrics are computed in the RGB channel, except for the image deraining task, which is evaluated in the Y-channel in YCbCr space, following previous works [23, 27].

*4.1.2 Implementation Details.* Our model consists of 20 CTLs while the channel number is set to 48. For image super-resolution, we train the model using 500K iterations with a patch size of 64 × 64 and a batch size of 32. The initial learning rate is set to 5 × $10^{-4}$ and is halved at the milestones: [250K, 400K, 450K, 475K]. For other restoration tasks, we extract patches of size 128 from

**Table 1: Quantitative comparison (PSNR/SSIM) of different lightweight models on benchmark datasets for image SR task. The best and second-best results are colored in red and blue, respectively. Note that CAMxierSR [41] only reports quantitative results at the ×4 scale.**

| Methods | Scale | Params | Set5 [5] PSNR/SSIM | Set14 [48] PSNR/SSIM | BSD100 [31] PSNR/SSIM | Urban100 [22] PSNR/SSIM | Manga109 [32] PSNR/SSIM | Average PSNR/SSIM |
|---|---|---|---|---|---|---|---|---|
| LatticeNet [29] | ×2 | 756K | 38.15/0.9610 | 33.78/0.9193 | 32.25/0.9005 | 32.43/0.9302 | 38.94/0.9774 | 35.03/0.9371 |
| SMSR [38] | ×2 | 985K | 38.00/0.9601 | 33.64/0.9179 | 32.17/0.8990 | 32.19/0.9284 | 38.76/0.9771 | 34.95/0.9365 |
| SwinIR-light [26] | ×2 | 878K | 38.14/0.9611 | 33.86/0.9206 | 32.31/0.9012 | 32.76/0.9340 | 39.12/0.9783 | 35.24/0.9390 |
| ESRT [28] | ×2 | 777K | 38.03/0.9600 | 33.75/0.9184 | 32.25/0.9001 | 32.58/0.9318 | 39.12/0.9774 | 35.15/0.9375 |
| FMEN [16] | ×2 | 748K | 38.10/0.9609 | 33.75/0.9192 | 32.26/0.9007 | 32.41/0.9311 | 38.95/0.9778 | 35.09/0.9379 |
| ELAN-light [59] | ×2 | 582K | 38.17/0.9611 | 33.94/0.9207 | 32.30/0.9012 | 32.76/0.9340 | 39.11/0.9782 | 35.26/0.9390 |
| DiVANet [4] | ×2 | 902K | 38.16/0.9612 | 33.80/0.9195 | 32.29/0.9012 | 32.60/0.9325 | 39.08/0.9775 | 35.19/0.9384 |
| NGswin [11] | ×2 | 998K | 38.05/0.9610 | 33.79/0.9199 | 32.27/0.9008 | 32.53/0.9324 | 38.97/0.9777 | 35.12/0.9384 |
| SRFormer-light [65] | ×2 | 853K | 38.23/0.9613 | 33.94/0.9209 | 32.36/0.9019 | 32.91/0.9353 | 39.28/0.9785 | 35.34/0.9396 |
| **Compacter (Ours)** | ×2 | 393K | 38.24/0.9619 | 34.08/0.9215 | 32.35/0.9026 | 32.99/0.9354 | 39.40/0.9786 | 35.41/0.9400 |
| LatticeNet [29] | ×3 | 765K | 34.53/0.9281 | 30.39/0.8424 | 29.15/0.8059 | 28.33/0.8538 | 33.63/0.9442 | 31.13/0.8738 |
| SMSR [38] | ×3 | 993K | 34.40/0.9270 | 30.33/0.8412 | 29.10/0.8050 | 28.25/0.8536 | 33.68/0.9445 | 31.15/0.8743 |
| SwinIR-light [26] | ×3 | 886K | 34.62/0.9289 | 30.54/0.8463 | 29.20/0.8082 | 28.66/0.8624 | 33.98/0.9478 | 31.40/0.8787 |
| ESRT [28] | ×3 | 770K | 34.42/0.9268 | 30.43/0.8433 | 29.15/0.8063 | 28.46/0.8574 | 33.95/0.9455 | 31.28/0.8759 |
| FMEN [16] | ×3 | 757K | 34.45/0.9275 | 30.40/0.8435 | 29.17/0.8063 | 28.33/0.8562 | 33.86/0.9462 | 31.24/0.8759 |
| ELAN-light [59] | ×3 | 590K | 34.61/0.9288 | 30.55/0.8463 | 29.21/0.8081 | 28.69/0.8624 | 34.00/0.9478 | 31.41/0.8787 |
| DiVANet [4] | ×3 | 949K | 34.60/0.9285 | 30.47/0.8447 | 29.19/0.8073 | 28.58/0.8603 | 33.94/0.9468 | 31.36/0.8775 |
| NGswin [11] | ×3 | 1,007K | 34.52/0.9282 | 30.53/0.8456 | 29.19/0.8078 | 28.52/0.8603 | 33.89/0.9470 | 31.33/0.8778 |
| SRFormer-light [65] | ×3 | 861K | 34.67/0.9296 | 30.57/0.8469 | 29.26/0.8099 | 28.81/0.8655 | 34.19/0.9489 | 31.50/0.8802 |
| **Compacter (Ours)** | ×3 | 399K | 34.69/0.9300 | 30.65/0.8477 | 29.28/0.8117 | 28.90/0.8659 | 34.31/0.9487 | 31.56/0.8808 |
| LatticeNet [29] | ×4 | 777K | 32.30/0.8962 | 28.68/0.7830 | 27.62/0.7367 | 26.25/0.7873 | 30.54/0.9075 | 29.01/0.8206 |
| SMSR [38] | ×4 | 1,006K | 32.12/0.8932 | 28.55/0.7808 | 27.55/0.7351 | 26.11/0.7868 | 30.54/0.9085 | 28.97/0.8209 |
| SwinIR-light [26] | ×4 | 897K | 32.44/0.8976 | 28.77/0.7858 | 27.69/0.7406 | 26.47/0.7980 | 30.92/0.9151 | 29.26/0.8274 |
| ESRT [28] | ×4 | 751K | 32.19/0.8947 | 28.69/0.7833 | 27.69/0.7379 | 26.39/0.7962 | 30.75/0.9100 | 29.14/0.8244 |
| FMEN [16] | ×4 | 769K | 32.24/0.8955 | 28.70/0.7839 | 27.63/0.7379 | 26.28/0.7908 | 30.70/0.9107 | 29.11/0.8238 |
| ELAN-light [59] | ×4 | 601K | 32.43/0.8975 | 28.78/0.7858 | 27.69/0.7406 | 26.54/0.7982 | 30.92/0.9150 | 29.27/0.8274 |
| DiVANet [4] | ×4 | 939K | 32.41/0.8973 | 28.70/0.7844 | 27.65/0.7391 | 26.42/0.7958 | 30.73/0.9119 | 29.18/0.8257 |
| NGswin [11] | ×4 | 1,019K | 32.33/0.8963 | 28.78/0.7859 | 27.66/0.7396 | 26.45/0.7963 | 30.80/0.9128 | 29.20/0.8262 |
| SRFormer-light [65] | ×4 | 873K | 32.51/0.8988 | 28.82/0.7872 | 27.73/0.7422 | 26.67/0.8032 | 31.17/0.9165 | 29.38/0.8296 |
| CAMixerSR [41] | ×4 | 765K | 32.51/0.8988 | 28.82/0.7870 | 27.72/0.7416 | 26.63/0.8012 | 31.18/0.9166 | 29.37/0.8290 |
| **Compacter (Ours)** | ×4 | 408K | 32.53/0.8994 | 28.88/0.7876 | 27.76/0.7440 | 26.69/0.8025 | 31.24/0.9148 | 29.42/0.8297 |

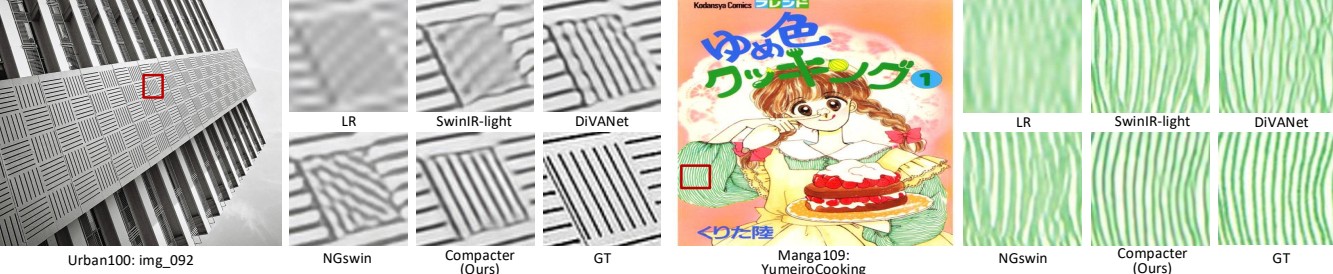

**Figure 5: Qualitative comparison for ×4 super-resolution on the Urban100 and Manga109 datasets.**

the training images with the batch size set to 32. We use 400K iterations to train the model. The initial learning rate is $1 \times 10^{-3}$ and steadily decreases to $10^{-7}$ as the cosine annealing decays. Following

[54], we use common horizontal and vertical flips, and random rotations of 90, 180, and 270 degrees for data augmentation. Note that no other data augmentation (e.g., Mixup, RGB channel shuffle)

**Table 2: Quantitative comparison on benchmark datasets for blind image denoising. $\sigma$ refers to the noise level.**

| Methods | Params | $\sigma$ | CBSD68 [31] PSNR | CBSD68 [31] SSIM | Kodak24 [18] PSNR | Kodak24 [18] SSIM | McMaster [58] PSNR | McMaster [58] SSIM | Urban100 [22] PSNR | Urban100 [22] SSIM |
|---|---|---|---|---|---|---|---|---|---|---|
| SwinIR-light [26] | 905K | | 34.16 | 0.9323 | 35.18 | 0.9269 | 35.23 | 0.9295 | 34.59 | 0.9478 |
| Restormer-light [53] | 1,054K | | 33.99 | 0.9311 | 34.86 | 0.9244 | 34.69 | 0.9229 | 34.00 | 0.9439 |
| CAT-light [8] | 1,042K | | 34.01 | 0.9304 | 34.90 | 0.9237 | 34.83 | 0.9247 | 34.12 | 0.9443 |
| ART-light [57] | 1,084K | 15 | 34.08 | 0.9315 | 35.00 | 0.9251 | 35.10 | 0.9282 | 34.44 | 0.9467 |
| NGswin [11] | 993K | | 34.12 | 0.9324 | 35.12 | 0.9268 | 35.17 | 0.9294 | 34.53 | 0.9476 |
| RAMiT-1 [10] | 818K | | 34.16 | 0.9324 | 35.13 | 0.9264 | 35.22 | 0.9297 | 34.58 | 0.9478 |
| **Compacter (Ours)** | 407K | | 34.22 | 0.9674 | 35.17 | 0.9674 | 35.30 | 0.9758 | 34.67 | 0.9726 |
| SwinIR-light [26] | 905K | | 31.50 | 0.8883 | 32.69 | 0.8868 | 32.90 | 0.8977 | 32.23 | 0.9222 |
| Restormer-light [53] | 1,054K | | 31.33 | 0.8865 | 32.38 | 0.8833 | 32.44 | 0.8905 | 31.60 | 0.9161 |
| CAT-light [8] | 1,042K | | 31.37 | 0.8855 | 32.43 | 0.8822 | 32.58 | 0.8928 | 31.75 | 0.9167 |
| ART-light [57] | 1,084K | 25 | 31.40 | 0.8864 | 32.49 | 0.8833 | 32.74 | 0.8956 | 32.03 | 0.9195 |
| NGswin [11] | 993K | | 31.44 | 0.8884 | 32.61 | 0.8865 | 32.82 | 0.8978 | 32.13 | 0.9215 |
| RAMiT-1 [10] | 818K | | 31.50 | 0.8888 | 32.64 | 0.8862 | 32.91 | 0.8989 | 32.21 | 0.9223 |
| **Compacter (Ours)** | 407K | | 31.57 | 0.9443 | 32.69 | 0.9478 | 32.97 | 0.9619 | 32.32 | 0.9569 |
| SwinIR-light [26] | 905K | | 28.22 | 0.8006 | 29.54 | 0.8089 | 29.71 | 0.8339 | 28.89 | 0.8658 |
| Restormer-light [53] | 1,054K | | 28.04 | 0.7974 | 29.19 | 0.8034 | 29.31 | 0.8256 | 28.30 | 0.8559 |
| CAT-light [8] | 1,042K | | 28.11 | 0.7960 | 29.29 | 0.8024 | 29.48 | 0.8296 | 28.46 | 0.8573 |
| ART-light [57] | 1,084K | 50 | 28.08 | 0.7950 | 29.27 | 0.8000 | 29.48 | 0.8279 | 28.62 | 0.8584 |
| NGswin [11] | 993K | | 28.13 | 0.8011 | 29.42 | 0.8087 | 29.59 | 0.8339 | 28.75 | 0.8644 |
| RAMiT-1 [10] | 818K | | 28.24 | 0.8024 | 29.51 | 0.8083 | 29.74 | 0.8376 | 28.93 | 0.8671 |
| **Compacter (Ours)** | 407K | | 28.36 | 0.8956 | 29.58 | 0.9066 | 29.78 | 0.9310 | 29.03 | 0.9199 |

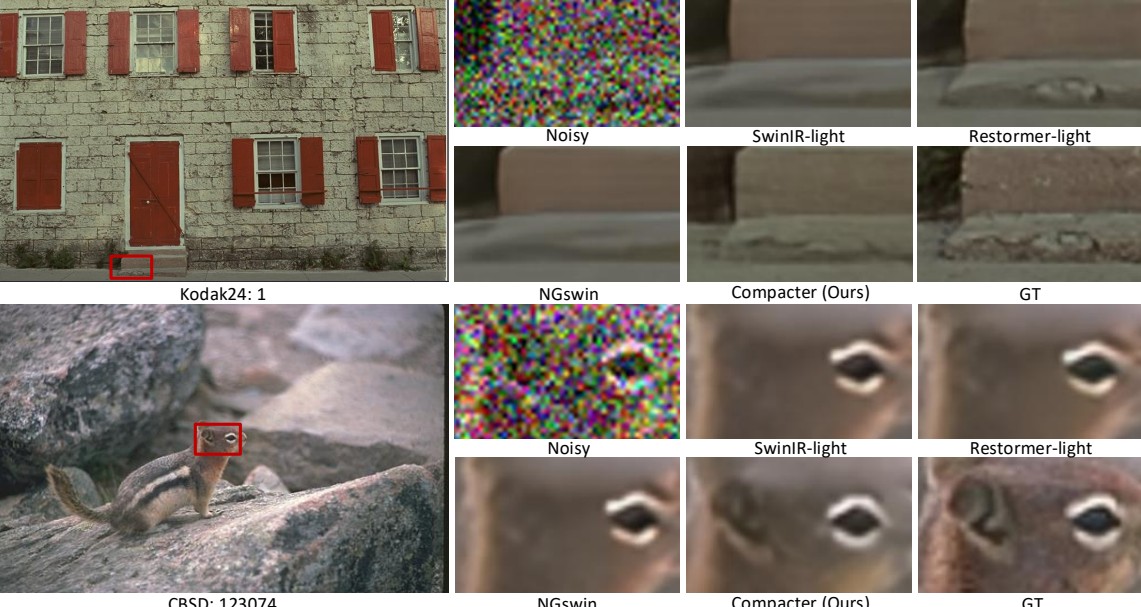

**Figure 6: Image denoising results with noise level $\sigma$ = 50 on the Kodak24 and CBSD datasets.**

or training skills (e.g., pre-training, multi-stage training) are used. The Adam algorithm was used with $\beta_1$ = 0.9 and $\beta_2$ = 0.99 for model optimization. All experiments are implemented in the PyTorch framework with four NVIDIA Tesla V100s.

## 4.2  Image Super-Resolution

We compare our method with state-of-the-art lightweight super-resolution algorithms, as shown in Table 1. It can be seen that our

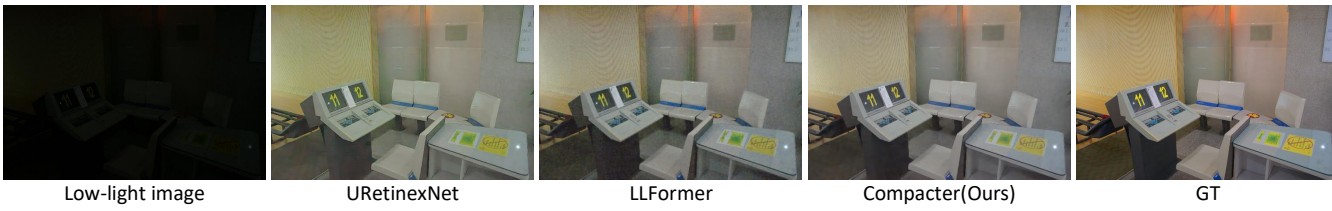

Low-light image | URetinexNet | LLFormer | Compacter(Ours) | GT

**Figure 7: Qualitative comparison on the LoL dataset for low-light enhancement. Please zoom in for a better view.**

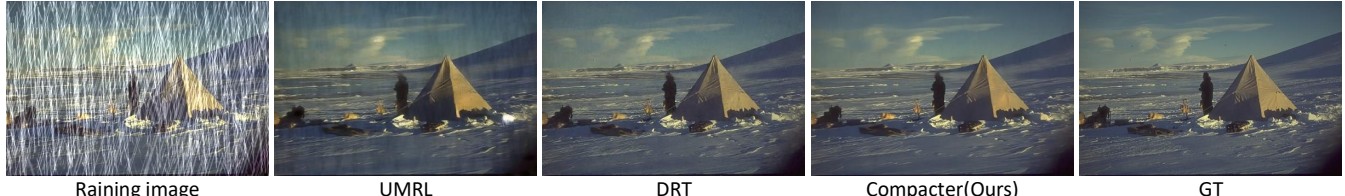

Raining image | UMRL | DRT | Compacter(Ours) | GT

**Figure 8: Qualitative comparison on the Rain100H dataset for image deraining. Please zoom in for a better view.**

**Table 3: Quantitative comparison for Low-light image enhancement on the LoL [44] dataset.**

| Method | Params | PSNR | SSIM |
|---|---|---|---|
| KinD [63] | 8,020K | 20.86 | 0.7900 |
| FIDE [47] | 8,620K | 18.27 | 0.6550 |
| DRBN [51] | 5,270K | 20.13 | 0.8300 |
| KinD++ [60] | 8,275K | 21.80 | 0.8338 |
| EnlightenGAN [24] | 8,640K | 17.48 | 0.6507 |
| Uformer [42] | 5,290K | 16.36 | 0.7110 |
| Restormer [53] | 26,130K | 22.43 | 0.8230 |
| URetinex-Net [45] | 1,230K | 21.32 | 0.8348 |
| KinD-SKF[46] | 8,500K | 21.91 | 0.8350 |
| LLFormer [39] | 24,520K | 23.65 | 0.8160 |
| **Compacter (Ours)** | 407K | 23.76 | 0.8375 |

**Table 4: Quantitative comparison for image deraining.**

| Method | Params | Test100 [56] PSNR/SSIM | Rain100H [50] PSNR/SSIM |
|---|---|---|---|
| UMRL [52] | 984K | 24.41/0.8290 | 26.01/0.8320 |
| JORDER-E [49] | 4,170K | 27.08/0.8720 | 24.54/0.8020 |
| MSPFN [23] | 13,350K | 27.50/0.8760 | 28.66/0.8600 |
| DRT [27] | 1,180K | 27.02/0.8470 | 29.47/0.8460 |
| TAO-Net [25] | 755K | 28.59/0.8870 | 28.96/0.8640 |
| RAMiT [10] | 935K | 30.44/0.9012 | 29.69/0.8775 |
| **Compacter (Ours)** | 407K | 30.63/0.9075 | 29.73/0.8826 |

### 4.3 Image Denoising

We follow [10, 53] to perform *blind* denoising experiments on the synthetic benchmark datasets generated using additive white Gaussian noise. The lightweight denoising methods used for comparison are derived from [9]. Table 2 shows the performances of different approaches on several benchmark datasets for noise levels 15, 25, and 50. It can be seen that our method achieves the best trade-off between efficiency and performance. Specifically, our method embraces significantly fewer parameters while obtaining superior performance compared with other methods. For example, for the challenging noise level 50 on the Urban100 dataset, our method outperforms SwinIR by 0.14 dB in PSNR, while the parameters are less than half of it. In addition, the far more perceptually relevant SSIM index shows that our Compacter has a significant advantage over other methods.

Figure 6 shows the denoising results of different methods. It can be seen that existing methods fail to recover enough details due to severe noise degradation. In contrast, our method provides better restoration of structure and detail, resulting in a clearer restoration.

### 4.4 Low-light Image Enhancement

We conduct low-light enhancement experiments on the LOL dataset. Table 3 shows the quantitative results, from which one can observe

Compacter achieves the best performance on almost all five benchmark datasets for all scale factors. In particular, our method achieves an average of 0.05 dB higher PSNR than the latest CAMixerSR (CVPR2024) with 46% fewer parameters. In addition, our method outperforms prominent SRFormer-light (ICCV2023) on average at all scales with less than half of the parameters. Compared to NGswin (CVPR2023), our method achieves up to 0.43 dB PSNR improvement on Manga109 with fewer parameters. These results demonstrate the superiority of our method.

In Figure 5, we also provide a visualization comparison of different lightweight SR methods at ×4 scale. It can be seen that competing algorithms are prone to erroneous textures and patterns. In contrast, the images produced by our method are clearer and visually closer to the ground truth. These visual results also illustrate the advantages of our Compacter in recovering details and textures.

that our method achieves better performance while having significantly fewer parameters. Specifically, our Compacter outperforms the acclaimed general-purpose image restoration method Restormer [53] by 1.33dB PSNR while requiring only 1.5% of its parameters. Furthermore, our method also outperforms the much larger LL-Former [39] and KinD-SKF [46], even though they are specifically tailored for low-light enhancement tasks. We further compared the visual results of the different methods in Figure 7. Compared to competitors, our method generates images with more natural and vivid colors.

## 4.5 Image Deraining

We further evaluate our method on the image deraining task. The qualitative results can be viewed in Table 4. As may be seen, our model improved over the state-of-the-art performances on both datasets. The average PSNR gain of our model over the latest lightweight model RAMiT [10] is 0.11 dB. Figure 8 shows a visual comparison on the more challenging Rain100H. Compared to other algorithms, our model produces images that are clearer and more consistent with the ground truth.

## 4.6 Ablation Study

For ablation experiments, we train super-resolution models on DIV2K for 200K iterations. The evaluations were performed on the Manga109 dataset [32] for ×2 scale super-resolution. FLOPs are calculated on input image size $3 \times 256 \times 256$.

**Compact Transformer Layer.** We performed ablation experiments to validate their effectiveness by removing CASA or DSGM from the CTL. For a fair comparison, we adjusted the depth of the models so that their parameters and FLOPs were comparable. Table 5 shows that removing CASA leads to severe performance degradation. This illustrates that the ability to model dense interactions is critical for restoration tasks that require per-pixel prediction. In addition, the absence of DSGM also leads to significant performance degradation. DSGM further improves the restoration performance by enhancing the useful information and suppressing the detrimental pixels through dual selective gating. In contrast, the full model achieves the best performance by coupling two complementary components.

**Compact Adaptive Self-Attention (CASA).** We conduct an ablation study to investigate the effectiveness of each component of CASA. As shown in Table 6, neither a single SWSA nor CWSA can achieve the desired performance, which suggests that uni-dimensional self-attention cannot realize comprehensive information interaction. Although the combination of the two attention techniques can achieve better performance, the absence of the *projection sharing* strategy leads to noticeable additional parameters and computational overheads. In addition, *adaptive interaction* further boosts performance barely costing any parameters and FLOPs. *Heterogeneous aggregation* also brings a 0.05dB performance improvement at a cost of a limited computational budget.

**Dual Selective Gated Module (DSGM).** We further explored the effectiveness of DSGM design as shown in Table 7. It can be seen that introducing depth-wise convolution in vanilla FFNs can bring performance improvement with few parameters and FLOPs. This strategy is also widely adopted by advanced image restoration

**Table 5: Ablation study of the CTL on the Manga109 dataset for ×2 scale image super-resolution.**

| Methods | PSNR | Params | FLOPs |
|---------|------|--------|-------|
| w/o CASA | 38.54 | 749K | 48G |
| w/o DSGM | 38.74 | 396K | 57G |
| Full Model | 39.29 | 393K | 47G |

**Table 6: Ablation study on the micro design of the CASA. *w/o projection sharing* denotes SWSA and CWSA independently use linear layer to generate $Q, K, V$. *w/o heterogeneous aggregation* means removing depth-wise convolution on the $V$.**

| Methods | PSNR | Params | FLOPs |
|---------|------|--------|-------|
| only SWSA | 39.24 | 393K | 42G |
| only CWSA | 38.98 | 393K | 30G |
| w/o projection sharing | 39.28 | 458K | 51G |
| w/o adaptive interaction | 39.25 | 393K | 47G |
| w/o heterogeneous aggregation | 39.24 | 384K | 46G |
| CASA (Ours) | 39.29 | 393K | 47G |

**Table 7: Ablation study on the micro design of the DSGM.**

| Methods | PSNR | Params | FLOPs |
|---------|------|--------|-------|
| FFN (Baseline) | 39.23 | 418K | 49G |
| ConvFFN (+DwConv) [65] | 39.28 | 435K | 51G |
| GDFN (+Gating Mechanism) [53] | 39.26 | 390K | 47G |
| DSGM (Ours) | 39.29 | 393K | 47G |

networks [42, 53]. In addition, the gating mechanism helps to further reduce the model parameters and computation but leads to some performance degradation. In contrast, our DSGM utilizing the dual selective gating mechanism can achieve superior performance while barely imposing additional computations.

## 5 Conclusion

In this paper, we present a novel compact Transformer for lightweight image restoration. Specifically, we propose compact adaptive self-attention (CASA) for comprehensive information dissemination and interaction within a compact computational unit. CASA simultaneously establishes the global context of spatial and channel as well as their interactions and achieves global-local coupling through heterogeneous aggregation. In addition, we propose a dual selective gated module (DSGM) to achieve dynamic context aggregation through a dual-path structure and gating mechanism. Thanks to two complementary components, our Compacter enables comprehensive pixel-level relational interactions while maintaining desirable model sizes. Extensive experiments demonstrate that Compacter achieves state-of-the-art performance on several restoration tasks with fewer model parameters.

# Acknowledgments

This work was partially supported by the National Natural Science Foundation of China under Grant 62072185, U1711262, 62173186, and 61703096.

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
