# OpenReview forum: "Compacter: A Lightweight Transformer for Image Restoration"
_acmmm.org/ACMMM/2024/Conference — MM2024 Poster_

### Official Review · Reviewer_WZh9 · 2024-05-20

**Rating:** 3
**Confidence:** 3

**Summary:**

In this paper, authors propose a compact Transformer (Compacter) for lightweight image restoration. In Compacter, they design Compact Adaptive Self-Attention (CASA) and dual selective gated network (DSGN). Thanks to these two complementary components, the Compacter achieves state-of-the-art performance on several restoration tasks with fewer model parameters.

**Strengths:**

The paper addresses the challenges of computational cost overhead in image restoration. Extensive experiments demonstrate that the proposed Compacter achieves SOTA results on various lightweight restoration tasks with significantly fewer parameters.

**Limitations:**

1）The proposed network in this paper builds upon existing CNN-based and Transformer-based approaches for image restoration. The lack of novelty in the paper stems from the fact that the proposed CASA and DSGN do not introduce fundamentally new concepts or techniques.

2）The name of Dual Selective Gated Network should be changed to improve the readability of the article, as it is only a key component of the Compact network. For example, it can be modified to Dual Selective Gated Block or Dual Selective Gated Module.

3）In the experimental section, it is recommended to add some inference speed experiments.

4）In the experimental section, some visual feature maps are suggested to be added to verify the effectiveness of the proposed module.

**Suitability:**

3

---

### Official Review · Reviewer_VUrn · 2024-05-27

**Rating:** 3
**Confidence:** 4

**Summary:**

This work proposes Compacter for lightweight image restoration by using projection sharing, adaptive interaction, and heterogeneous aggregation. Compacter achieves state-of-the-art performance for a variety of lightweight IR tasks.

**Strengths:**

1. The paper is easy to follow.
2. The results seems promising.

**Limitations:**

1. Design of the network is not novel.  I don’t see much new stuff, most of it is just existing work being integrated together.
2. The paper lacks runtime comparison for all experiments. Only the parameters and Flops comparison is not enough.
3. More results on Real-world denoising, super-resolution and image dehazing is necessary.

**Suitability:**

2

---

### Official Review · Reviewer_f8u5 · 2024-05-27

**Rating:** 4
**Confidence:** 2

**Summary:**

This paper proposes a novel approach to image restoration with Compacter, a compact and efficient Transformer-based model. The key innovations include the Compact Adaptive Self-Attention (CASA) mechanism and the Dual Selective Gated Network (DSGN). CASA integrates spatial and channel self-attention using shared projections, enhancing global-local feature interactions. DSGN dynamically adjusts and aggregates features through parallel branches and gating mechanisms, improving restoration quality. With significantly fewer parameters, Compacter still achieves competitive or better performance across various image restoration tasks against state-of-the-art methods.

**Strengths:**

- The proposed Compacter achieves impressive results on multiple restoration tasks and the authors provide extensive experimental results to demonstrate the effectiveness.

- The use of projection sharing and efficient attention mechanisms significantly reduces the number of parameters while outperforming SOTA methods.

- The whole structure is straightforward to follow and easy to modify for different tasks.

**Limitations:**

- Although the experiments show that the model is effective, I think that the authors need to go further and demonstrate why the design of each component is valid. Especially for Tab 6, with only SWSA and 393K parameters, the model can already outperform most of the baseline models. However, in Tab 5, removing CASA leads to severe performance degradation with 749K parameters. I suggest that the authors provide further explanation of this.

- I am confused about some reported results:
    1) For the ablation study, the PSNR of the full model is 39.29 (Tab 5, 6, 7). But in Tab 1 it's 39.40. Why are they different?
    2) For Tab 6, with SWSA or CWSA, the number of parameters does not decline significantly. Why this is the case? If the authors "adjusted the depth of the models so that their parameters and FLOPs were comparable" as in Tab 5, I think it needs to be declared.
    3) For Tab 5, how do authors implement "w/o CASA"?

- The authors didn't illustrate or cite how the comparison methods (ConvFFN and GDFN) are implemented. Also, the results cannot explain why the design of DSGN is effective, i.e. why choose GELU and Sigmoid?

- In Fig 3, Q, K, and V should come from three different linear layers rather than one linear layer (see Eq. 5).

- In Fig 2 b), there are two "element additions" missing.

**Suitability:**

2

---

### Meta-Review · Area_Chair_5bEN · 2024-07-01

**Recommendation:** Accept (Poster)
**Confidence:** 5

**Metareview:**

The paper received (2) borderline accept and (1) borderline reject. However, the negative comments are too general, which are not the key limitations of the proposed method. This paper presented a compact and efficient transformer-based model for image restoration. The motivation is clear, and all reviewers recognized the contribution of the reasonable design of the architecture, and the experimental results are sufficient to support the lightweight model and achieve better performance. We congratulate the authors on the acceptance of their paper!